# The Formulation and Evaluation of Deep Eutectic Vehicles for the Topical Delivery of Azelaic Acid for Acne Treatment

**DOI:** 10.3390/molecules28196927

**Published:** 2023-10-04

**Authors:** Dhari K. Luhaibi, Hiba H. Mohammed Ali, Israa Al-Ani, Naeem Shalan, Faisal Al-Akayleh, Mayyas Al-Remawi, Jehad Nasereddin, Nidal A. Qinna, Isi Al-Adham, Mai Khanfar

**Affiliations:** 1Faculty of Pharmacy, Pharmacological and Diagnostic Research Center, Al-Ahliyya Amman University, Amman 19328, Jordan; dluhaibi@amman.edu.jo (D.K.L.); n.shalan@ammanu.edu.jo (N.S.); 2Department of Pharmaceutics, College of Pharmacy, University of Sulaimani, Sulaimani 46001, Kurdistan Region, Iraq; hiba.mohammed@univsul.edu.iq; 3Faculty of Pharmacy and Medical Sciences, University of Petra, Amman 11196, Jordan; falakyleh@uop.edu.jo (F.A.-A.); malremawi@uop.edu.jo (M.A.-R.); ialadham@uop.edu.jo (N.A.Q.); nqanna@uop.edu.jo (I.A.-A.); 4Faculty of Pharmacy, Department of Pharmaceutical Sciences, Zarqa University, Zarqa 13110, Jordan; jnasereddin@zu.edu.jo; 5Department of Pharmaceutical Technology, Faculty of Pharmacy, Jordan University of Science and Technology, Irbid 22110, Jordan; mskhanfar@just.edu.jo

**Keywords:** Azelaic acid, ionic liquid, deep eutectic mixture, choline chloride, malonic acid

## Abstract

The current work was aimed at the development of a topical drug delivery system for azelaic acid (AzA) for acne treatment. The systems tested for this purpose were deep eutectic systems (DESs) prepared from choline chloride (CC), malonic acid (MA), and PEG 400. Three CC to MA and eight different MA: CC: PEG400 ratios were tested. The physical appearance of the tested formulations ranged from solid and liquid to semisolid. Only those that showed liquid formulations of suitable viscosity were considered for further investigations. A eutectic mixture made from MA: CC: PEG400 1:1:6 (MCP 116) showed the best characteristics in terms of viscosity, contact angle, spreadability, partition coefficient, and in vitro diffusion. Moreover, the MCP116 showed close rheological properties to the commercially available market lead acne treatment product (Skinorin^®^). In addition, the formula showed synergistic antibacterial activity between the MA moiety of the DES and the AzA. In vitro diffusion studies using polyamide membranes demonstrated superior diffusion of MCP116 over the pure drug and the commercial product. No signs of skin irritation and edema were observed when MCP116 was applied to rabbit skin. Additionally, the MCP116 was found to be, physically and chemically, highly stable at 4, 25, and 40 °C for a one-month stability study.

## 1. Introduction

Acne (also called acne vulgaris (AV)) is a common dermatological disorder that most frequently affects adolescents, yet people from all age groups are candidates to be affected at least once at some point in life [1,2]. The visible nature of acne, symptoms, and sequelae all contribute physically and psychosocially to the overall burden of the disease, as do the costs required for management [3]. Sebum production, accumulation of dead skin cells in follicles, and hormonal factors are the endogenous factors that contribute to acne formation; however, the major cause of acne pathogenesis is the microorganisms [4]. Among these microorganisms, Gram-positive *Propionibacterium acnes* (*P. acnes*) has been the primary cause, whereas microorganisms such as *Staphylococcus aureus* (*S. aureus*) and *Staphylococcus epidermidis* (*S. epidermidis*) are known to increase the severity of this disease [5,6]. The widespread and prolonged use of antibiotics introduces a potential added burden through resulting antimicrobial resistance [7]. Amongst the most commonly used antibiotics, Azelaic acid (AzA, Figure 1) is a naturally occurring saturated dicarboxylic acid that has therapeutic relevance in the management of several dermatological conditions. AzA is reported to possess keratolytic and comedolytic properties, and is reported to be effective in the management of hyperpigmentation disorders like lentigo maligna and melasma [7,8,9]. AzA is also reported to possess antibacterial, specifically bactericidal, activity against *Cutibacterium acnes* and *Staphylococcus epidermidis*, and is therefore commonly used in the management of acne vulgaris [10,11,12].

AzA is regarded as a problematic drug to formulate; it is sparingly soluble in water, and is reported to possess limited skin permeability [13]. Several formulation strategies have been investigated to facilitate the intradermal delivery of AzA, including microemulsions [14], ethosomes [15], liposomes [16], foam formulations [17], hydrogels [18], and chitosan-loaded nanoparticles [19], among others.

Of the various strategies to enhance drug solubility and skin penetrability, Ionic Liquids (ILs) and deep eutectic systems (DESs) have recently gained considerable interest as green solvents with unique tunable physiochemical properties, most interestingly their superior solvation properties, wide liquid ranges, safety, non-toxicity, non-flammability, non-volatility, thermal stability, sustainability, biodegradability, skin penetration enhancement, antibacterial activity, and low cost [20,21,22]. Plenty of recently published research showed the potential antibacterial activity of some DESs [23]. It was reported that choline chloride (CC, Figure 1) with urea or glucose DESs did not inhibit bacterial growth, while the DES made from CC and malonic acid (MA, Figure 1) had an inhibitory effect, indicating its potential application as an antibacterial agent [24] (Marcel et al., 2022). Recently, DES made from choline and geranic acid has been shown to exhibit excellent antimicrobial activity against a broad variety of pathogens [25]. Further, these DESs have been shown to exhibit deep penetration into the skin, thus suggesting their ability to treat pathogens residing in the skin [26].

From the literature, DESs based on CC and dicarboxylic acids (such as MA) are one of the most commonly used representatives of these fluids that at certain molar ratios are room-temperature liquid low-melting mixtures [24,27]. One of the critical drawbacks of DESs is their relatively higher viscosity compared to molecular solvents [28]. The high viscosity of these DESs limits their pharmaceutical applicability as solubilizers and drug vehicles. To overcome their high viscosity, one efficient way is to use an appropriate co-solvent such as propylene glycol, polyethylene glycol, or glycerol. Moreover, the suggested components of the DES utilized in the current study have been reported as suitable vehicles for topical applications and are generally recognized as safe (GRAS) when used in accordance with good manufacturing practices [29]. To the best of our knowledge, the pharmaceutical application of a DES made from MA and CC as a solubilizing and drug delivery vehicle for AzA was not reported before. Therefore, the present work describes an investigation into the feasibility of the use of a DES formulation consisting of MA, CC, and PEG 400 as a vehicle for the intradermal delivery of AzA to treat acne. The prepared DES will be evaluated in terms of viscosity, spreadability, pH, partition coefficient, contact angle, and solubilization power.

## 2. Results and Discussion

### 2.1. Rheological, Solubility, and Contact Angle Studies

The acceptability and efficacy of topically applied products require that they have optimal mechanical properties, adequate rheological behavior, spreadability, and appropriate skin adhesion. Rheological evaluation is mainly to ensure the suitability of product manufacturing, uniformity, extrudability, stability, and suitability with respect to rubbing on the human skin. The rheological study (Figure 2) was conducted at temperatures that could represent product storage in the refrigerator (5 to approximately 10 °C), room temperature (22 to 25 °C), and skin temperature (32 °C). For temperatures in the range from 8 to 20 °C, the order of viscosity was MC21 > MC12 > MC11. The extremely high viscosity of MC21 hinders its application as a suitable solvent and drug carrier system, and it was therefore excluded from further investigations [30]. At room temperature (22 to 25 °C) and skin temperature (32 °C), formulation MC11 showed significantly lower viscosity readings than MC12. To further decrease the viscosity, PEG 400 was added as a co-solvent at different ratios to MC11 and MC12 and was compared with a reference commercial topical product for acne treatment (Skinoren^®^, 20% AzA, Leo laboratoreies Ltd., Maidenhead, UK). PEG 400 addition resulted in a significant reduction in the viscosity readings of the DESs. The MCP116 showed the lowest viscosity reading among the tested DESs (Table 1). MCP116 viscosity value was closer to that of the commercial product (340 ± 7.1 MPa and 380 ± 5.1 MPa for MCP116 and Skinoren^®^, respectively). PEG400 has hydrophilic polymer chains with the availability of many terminal hydroxyl groups that may possibly improve the hydrophilicity of biomaterials. In the current study, the addition of PEG 400 to MC11 and MC12 resulted in an increase in the partition coefficient parallel with a decrease in the contact angle on the hydrophobic glass surface. The increased hydrophobicity with increased PEG addition suggests possible interruptions of MA and CC interactions. In addition, PEG400 is less polar as compared to ionic liquids and DESs [31]. Therefore, the PEG addition resulted in a significant decrease in the DES contact angles, with MCP116 being the lowest. The low contact angle of MCP116 indicates its good hydrophobicity. Although AzA contains hydrogen bond donors and acceptors, AzA (pKa = 4.6) in the ionized form has a small diffusivity through the stratum corneum [32]. The pH of the DESs (2.60 to 3.13, acidic pH) renders AzA mostly in the unionized form. Therefore, AzA alone and AzA in the DESs partitioning properties were evaluated. AzA’s apparent solubility in octanol was (45 ± 4) mg/mL at 22 °C. The estimated Octanol/water log *p* value of AzA was approximately 2 ± 0.02. The Octanol/water log *p* value of AzA is comparable to 1.42 ± 0.06 [33]. Increasing the ratio of PEG with resulted in an increase in log *p* with MCP116 being the highest. Such an increase in partitioning coefficient values could be due to the existence of the drug in the unionized form in the DESs system. Such results could be due to the availability of many (OH) groups for hydrogen bonding with the AzA and the decrease in the viscosity of the DESs with PEG addition. A decrease in viscosity, a reduction in contact angle, and an elevated partitioning coefficient collectively promote the spreadability of the formula, with MCP116 achieving the highest value, as indicated in Table 1. Additionally, the incorporation of PEG400 in the DESs was observed to have no impact on the pH. This outcome is to be expected, as PEG represents a nonionizable polymer, devoid of proton-donating or proton-accepting attributes. Most importantly, the results presented in Table 1 showed an increase in drug solubility in the DES formulations compared to that of pure water. Among the tested DESs, MCP116 showed the highest solubility, with about 80 times the solubility of the drug in water (AzA solubility in water is 2.4 mg/mL) [13]. The high-required concentration (20%) of AzA to guarantee its availability in the skin increases the incidence of side effects, most likely skin irritation. Moreover, such a high required drug concentration increases the need for a drug carrier system with high loading capacity, acceptable skin tolerability, and good penetration enhancement ability. DESs were shown to be a possible solution to overcome these challenges. The pH values of the formulations in the range of 2.6 to 3.11 are generally lower than the typical pH range of healthy skin, which falls between 4 and 6. Lower pH values can make formulations more acidic, which could potentially lead to skin irritation, dryness, or disrupt the skin’s natural pH balance. However, the suitability of these pH values depends on various factors, including the specific ingredients in the formulation, the intended use (e.g., as a spot treatment or all-over application), and individual skin sensitivities. Since the current formulation is intended for short-term and localized use (acne treatment), a slightly lower pH might be acceptable and justifiable as it can help solubilize AzA, and enhance its effectiveness and intradermal penetration. Furthermore, it is worth noting that the components comprising the formulation, namely CC, MA, and PEG400, are generally considered “green” components. They have a well-documented history in skin and cosmetic formulations, demonstrating acceptable levels of skin irritation and tolerability. These attributes can further support the justification for the pH range chosen in the formulation. Because of the high viscosity of PEG-free DESs (MC11, MC12, and MC21) solubility of AzA was not possible to perform.

### 2.2. Attenuated Total Reflectance—Fourier Transformed Infra-Red (ATR–FTIR)

FTIR studies were successfully implemented for the elucidation of the possible interactions between DES components [33]. The ATR-FTIR results of MA, CC, AzA, PEG, and the optimal candidate formulation MCP116 are shown in Figure 3. AzA exhibited characteristic peaks at 1706 cm^−1^, likely corresponding to the C=O stretching peak. The peak centered around 3000 cm^−1^ likely corresponds to the carboxylic acid dimer structure of AzA. The peaks observed between 2800 and 2950 cm^−1^ are among the specific peaks of the aliphatic chains of AzA [34]. CC exhibited characteristic peaks at 3391 cm^−1^, and 1086 cm^−1^, likely corresponding to the -OH stretching, and C-N stretching peaking, respectively. PEG exhibited a characteristic -OH stretching peak at 3374 cm^−1^. MA exhibited a characteristic, aggregated peak centered on 1695 cm^−1^, likely corresponding to C=O stretching; it also exhibited the carbonyl dimer ring centered on 3000 cm^−1^. The spectra of both the blank (placebo DES) and the drug-loaded DES (MCP116) appear nearly identical; the carbonyl dimer is seen shifted to 2900 cm^−1^, with the peak appearing broader in the drug-loaded MCP116 formulation. Furthermore, there is a clear shifting of the C=O stretching peak MA, indicative of hydrogen bonding. Both the peaks corresponding to a carbonyl functional groups and the -OH stretching peak are seen to be more intense (relative to % transmittance) in the drug-loaded formulation than in the placebo, but the characteristic peaks of AzA were not visible, likely due to the penetration depth limit of the ATR crystal.

### 2.3. Differential Scanning Calorimetry (DSC) Study

The solid materials analyzed (Figure 4) exhibited typical thermal behavior and melting endotherms at 105 °C, 151 °C and 325 °C for raw AzA, MA and CC, respectively (Figure 4). Such results are consistent with those reported in the literature [35,36]. Concerning PEG, a substance that remains in a liquid state at room temperature, thermal analysis was initiated at −40 °C when it was in a solid phase. The thermogram revealed an exothermic peak at −30 °C, signifying solidification, and an endothermic peak at 5 °C, indicating melting. Furthermore, Figure 4 illustrates the thermogram of MCP116 in combination with AzA. Notably, this thermogram displays a minor peak at −20 °C, another minor peak at 150 °C, and a broad peak at 260 °C. The findings indicated the absence of the characteristic peaks associated with MA and AzA, replaced by a broad peak at 260 °C, indicative of a lowered melting point for CC. These results substantiate the formation of a DES between MA and CC, the complete miscibility of PEG400, and the full solubilization of AzA within the resultant DES.

### 2.4. High-Performance Liquid Chromatography (HPLC)

A singular, sharply defined peak corresponding to AzA, displaying remarkable resolution, emerged at a retention time (RT) of 4.278 min, as depicted in Figure 5. The analytical method exhibited robust linearity, substantiated by a high correlation coefficient (R^2^ = 0.9997), and demonstrated good selectivity by virtue of the absence of any discernible interference from the excipients with the model drug.

### 2.5. Diffusion Study

Based on the results regarding partitioning coefficient, contact angle, spreadability, and rheology, MCP116 was chosen as the candidate DESs and MCP114 was tested for comparative purposes. To assess the permeation characteristics, equivalent quantities of AzA were investigated from both formulations, alongside a pure drug solution and a commercially available product, using a Franz diffusion cell. Previous studies established the successful use of polyamide membranes as an in vitro model for SC [37,38]. Therefore, the diffusion of AzA and the selected formulations through polyamide membranes was studied. The concentration of AzA in the donor phase was the same in all diffusion studies. As anticipated, due to its higher lipophobicity, the MCP116 formulation exhibited greater diffusion compared to the parent drug.

The findings depicted in Figure 6 indicate that all formulations exhibited comparable flux during the initial 5 h. However, the flux demonstrated a consistent linear augmentation, characterized by a high R^2^ value of 0.99, reaching values of 23.5 ± 4.1, 16.7 ± 2.0, and 7.5 ± 1.01 mg.cm^2^.h for MCP116, MCP114, and Skinorin^®^, respectively, after 24 h.

The cumulative amount released (as an average of three readings) was equal to approximately 100 ± 7%, 75 ± 8%, and 16.5 ± 3%, from MC116, MC114, and Skinorin^®^, respectively. These results showed the superiority of MC116 over MC114 and the reference product Skinorin^®^. The pure drug showed only low diffusion through the tested membrane.

### 2.6. Microbiological Study

Recently many reports indicated the potential application of organic acid-based MA: CC DESs as an antibacterial agent due to their inhibitory effect [23]. A recent study investigated the DES composed of a 2:1 ratio of MA and CC along with its individual component, MA, for their antifungal properties against notable fungal species, including *Aspergillus niger*, *Lentinus tigrinus*, *Candida cylindracea*, and *Cyprinus carpio fish*. The study’s findings concluded that the MA present in the DES exhibited lower toxicity when compared to its isolated counterpart, MA. [39]. AzA inhibits protein synthesis inside bacterial cells [40]. In the current study, the low pH offered by the DES (pH 3.13 ± 0.01), renders AzA mostly in the unionized form; thus, enhancing its uptake through the bacterial cell wall as also proposed by a study by Al-Marabeh et al. [41]. Table 2 showed an inhibitory effect on the growth of tested microorganisms with an inhibition zone of 28.62 ± 0.85 mm, 13.03 ± 0.90 mm, and 21.50 ± 0.81 mm for MCP116 with AzA, MCP116 without AzA (blank DES), and Skinorine^®^, respectively. Such results indicate that the drug alone showed an inhibitory effect higher than the blank DES. MCP116 with AzA showed a significantly higher inhibitory effect against *P. acnes*, mostly due to the additive effect between the drug and MA of the DES. While the effectiveness of the MCP116 with AzA compared to the marketed product was statistically significant (*p* < 0.05), no statistical difference was detected between the inhibition zones of MCP116 without AzA and the commercial product (*p* > 0.05). The relatively small inhibition zone diameter observed for MCP116 in the absence of AzA is likely attributable to MA, a phenomenon previously documented in several references [42,43]. This effect could be a result of the additive interaction between MA and AzA. Such formulation (i.e., MCP116 with AzA) exhibited higher efficacy than the marketed product, since they yielded a larger inhibition zone diameter, which might be explained by the fact that MCP116 with AzA scattered faster in the medium due to its high spreadability and diffusion characteristics.

The assessment of skin irritation or corrosive potential was based on the Draize Dermal Irritation Scoring model, following a methodology consistent with prior findings [44]. The results are summarized in Table 3, and skin responses were consistently observed throughout all stages of the dermal testing process.

The results obtained (zero erythema and zero edema) indicate that the application of the formula on rabbit skin, serving as a model for human skin, did not exhibit any signs of irritation for a duration of 72 h. This suggests that both the DES and the formula can be safely applied to the skin without a significant risk of irritation.

### 2.7. Stability Results

The physical appearances of all formulations at all tested storage conditions were unchanged in terms of phase separation and transparency, demonstrating that the AzA in the DES system is thermodynamic stable. Moreover, the drug did not show any sign of precipitation. After storage at 25 °C for 30 days, the AzA content was 98.2 ± 2.2% and the rheological behavior of the formulation did not change, indicating that the tested AzA-loaded in DES formulations were considered stable.

In summary, this formulation represents a pioneering approach to AzA delivery for acne treatment, utilizing DES. It offers several advantages, including safety, environmental friendliness (as it is free of organic solvents), and simplicity in preparation. Anti-acne studies demonstrated that the DES formulation exhibited inhibition zones comparable to those of the commercially available cream and previously reported formulations [15,34,45].

## 3. Materials and Methods

### 3.1. Materials

Azelaic acid was purchased from UFC Biotechnology, USA, Choline Chloride 99.2% and Malonic Acid 99.1% (Xi’an Gawen Biotechnology company (Xi’an, China), Sodium dihydrogen phosphate, Potassium dihydrogen phosphate and Phosphoric acid (Merck, Germany), Formic acid (HPLC grade), Hydrochloric acid 33% (HPLC grade), Polyethylene glycol 400 and Dimethyl sulfoxide (TEDIA, Fairfield, OH, USA), Acetonitrile, 99% (VWR chemicals, BDH^®^, Radnor, PA, USA), Dialysis tubing (Medical International Ltd., London, UK).

### 3.2. Preparation of Deep Eutectic Systems (DESs)

DES formulations consisting of different ratios of MA and CC, either with or without the addition of PEG 400, are presented in Table 4. One gram of each eutectic system was prepared by mixing the components in a glass bottle with continuous stirring at room temperature. The temperature was gradually increased from 25 °C to 100 °C, and stirring was continued until a clear liquid was achieved. Each mixture was then degassed using an Ultrasonic Cleaner (Elma Sonicator, Sign, Germany). Finally, each eutectic formulation was equilibrated in a shaking water bath (SB-12L Shaking Water Baths) for 12 h at 190 rpm and 25 °C.

### 3.3. pH Measurement of DESs

The pH of each of the prepared DESs was measured using a pH meter (Jenway 3510 pH meter). A total of 1 mL from each eutectic system was transferred into a beaker then diluted with distilled water up to 20 mL, and the pH was measured using a pH meter (Jenway 3510 Standard Digital pH Meter Kit; 230 VAC/UK). Each measurement was conducted in triplicate, and is reported as the mean ± standard deviation.

### 3.4. Solubility Study

The solubility of AzA in the DESs was measured by adding 200 mg of AzA to 2 gm of each of the DESs in small glass bottles. Each glass was then sealed and transferred to a vortex mixture for 15 min, and then equilibrated in a shaking water bath for 24 h at 190 RPM and 25 °C. Each sample was then centrifuged (Stuart SCF1 Mini Centrifuge Spinner) at 14,000× *g* RPM for 5 min. The supernatant was assayed for AzA present. The results are reported as the percent dissolved relative to the original 200 mg placed in each container. Solubility measurements were conducted for samples that showed good viscosity compared to water. Each measurement was conducted in triplicate, and is reported as the mean ± standard deviation.

### 3.5. High-Performance Liquid Chromatography (HPLC)

HPLC was used to quantify the presence of AzA in the DESs. A Supelcosil LC-18-DB (5 µm, 15 cm × 4.6 mm) column was used. The mobile phase consisted of a 75% Phosphate-Buffered Saline (PBS) at pH 3.5, and a 25% acetonitrile mixture at a flow rate of 1 mL/min. The injection volume was 20 µL, the column temperature was set to 40 °C, and the UV detector was set to 206 nm. The total run time was 10 min. The quantification of AzA was performed against a standard calibration curve that was previously prepared in the region of 22–280 µg/mL.

### 3.6. Attenuated Total Reflectance—Fourier Transformed Infra-Red (ATR-FTIR) Study

ATR-FTIR spectra of AzA, CC, MA, and DES MCP116 (both with and without the presence of the drug), were carried out using (Perkin Elmer Spectrum Two UATR FT-IR spectrometer, Waltham, MA, USA). Scan resolution was set to 4 cm^−1^, with 32 samples per scan. Spectra were acquired over a range of 4000–450 cm^−1^.

### 3.7. Differential Scanning Calorimetry (DSC) Study

DSC was performed on AzA, MA, CC, PEG 400, and MCP116. Each test was performed by weighing an approximate amount between 5 and 8 mg of solid samples and an amount of 20–30 mg of liquid samples into an aluminum DSC crucible. Analysis was performed using the Mettler Toledo DSC 1 Star system. The temperature limit for the samples was between −60 and 320 °C, depending on the sample. All measurements were performed at a heating rate of 10 °C/min.

### 3.8. Rheology Study

The viscosity of different prepared DESs was studied using a rheometer (Physica MCR 302, Anton Paar, Graz, Austria) that offers different geometries (concentric cylinder, cone-and-plate, and parallel-plate) with Cp 50 double gap concentric cylinder measurement system to determine the viscosity. After the calibration of the instrument, each DES was loaded between the concentric cylinders at a volume of approximately 5–10 mL. Measurement conditions were: shear rate (0.1–100), temperature set (−10–30 °C), cone angle 1, and zero-gap 0.1.

### 3.9. Measurement of Partition Coefficient

The partition coefficient of AzA was measured in the selected DES formulations. For measurement, 20 mL octanol was placed in a flask, to which 0.5 g of the sample to be tested was accurately weighed and added, followed by the addition of 30 mL of deionized water. The mixtures were then placed in a shaking water bath and mixed at 190 RPM and room temperature for 24 h. Each mixture was then transferred to a separatory funnel and allowed to equilibrate. The octanol was then carefully decanted using the separatory funnel. A total of 2 mL of each phase was then diluted with the mobile phase and assayed for drug content the HPLC method. The partition coefficients of the tested samples are reported as the ratio of octanol content to water content, respectively.

### 3.10. Spreadability

The spreadability of the DESs was investigated by placing 1 g of the DES preparation to the center of a 20 cm × 20 cm glass plate. It was then covered with an identical slide and waited for a minute. The diameter of the spread area (cm) of three triplicates was measured as the mean ± SD [46].

### 3.11. In Vitro Drug Release with Franz Diffusion Cells

AzA diffusion was investigated using standard Franz diffusion cells (SES GmbH-Analyse Systeme/Fridhofstr 7–9D 55234 Bechenheim/Germany), using a polyamide membrane, 0.2 µm pore size (Sartolon Polyamide, Germany) as the diffusion barrier with a cross-sectional area of 1.7 cm^2^. The acceptor compartment consisted of 12 mL Phosphate-Buffered Saline (PBS) pH 7.4 with 30% ethanol as a cosolvent to maintain sink condition (Arpa et al., 2022). The receptor cell was stirred at 550 rpm by a magnetic stirrer, and maintained at 37 °C by circulating water with a thermostatic pump. Pure AzA (200 mg) and a suitable amount of selected DES formulations, and a commercially available AzA formulation (equivalent to 200 mg AzA) were added to the donor compartment. A drug solution in PBS pH 7.4 with 30% ethanol was also tested as a control. Aliquots of 0.5 mL were withdrawn from the acceptor compartment and replaced with equal volumes of the respective PBS pH 7.4 with 30% ethanol for 24 h. The withdrawn samples were analyzed using an HPLC method.

### 3.12. Contact Angle Measurements

The contact angles of a drop of the DES were measured on a polyethylene plastic surface using the contact angle goniometer (OCA 15 EC, Data Physics instruments GmbH, Filderstadt, Germany) running SCA20 Software (https://sca20.software.informer.com/) for OCA and PCA. For each measurement 500 µg Hamilton syringe was filled with the sample and anchored on the device. The dosing volume of each drop was 4 µg, with a dosing rate of 1 µg/s. High-resolution images of each drop were captured by a fixed camera. Analysis of images was carried out by two baselines, a curve, and the tangent angle drawn by the software.

### 3.13. Microbiological Study

The microbiological activity of the selected formula (200 mg AzA in MCP116) against Propionibacterium acnes was tested. The sensitivity test to the formula was assessed by the disc diffusion method. AzA was taken as a positive control and the blank formula as a negative control. The concentration of the bacteria was 108 CFU, the temperature of incubation was 37 °C, and the type of media was blood agar and anaerobic jar. Four disks were put in each Petri dish soaked with the diluted sample of formula MCP116 (2 disks), blank formula MCP116 (without AzA), and AzA in the solvent. Dilutions were made by taking 400 mg of both MCP116 and blank MCP116, which were diluted in 0.5 mL DMSO, and 66.5 mg pure AzA was dissolved in 0.5 DMSO. A total of 20 µL of each sample was pipetted and put on the sterilized disk and the disk was put in the specified place on the Petri dish. The Petri dish was then put in the incubator for 72 h in anaerobic conditions.

### 3.14. Skin Irritation/Corrosive Potential Test

Healthy young albino female rabbits (2250 ± 150 g) were housed and acclimatized at the Laboratory Animal Research Unit of the University of Petra Pharmaceutical Center. All animal studies were conducted following the University of Petra Institutional Guidelines on Animal Use. Rabbits were individually housed in rabbit racks (X-type, Techniplast, Italy) and maintained under controlled conditions of temperature (20 ± 3 °C), humidity (50 ± 15%), and photoperiod cycles (12 light/12 h dark) with a conventional laboratory diet and unrestricted supply of drinking water. Assessment of skin irritation/corrosive potential and the reversibility of dermal effects of a topical preparation of AzA. The presented test was conducted according to the OECD Guideline for Testing of Chemicals, adopting Guideline 404 for Acute Dermal Irritation/Corrosion.

A dose of 0.5 mL of both the formula and vehicle was applied to an area of approximately 6 cm^2^ of the skin and covered with a gauze patch. According to the guideline an initial test, using one animal was conducted by applying three patches sequentially on the rabbit at different sites. Briefly, the first patch was removed after three min, then when no serious skin reaction was observed, a second patch was applied and removed after one hour. Thereafter, a third patch was applied and left for four hours only. Animals were examined immediately after patch removal for signs of erythema and edema, and dermal reactions were scored.

A confirmatory test was conducted after the initial test, since no corrosive effects were observed. Thus, the negative response was confirmed using another animal for an exposure period of 4 h. Then, the dermal response was evaluated immediately after the removal of the patch, after 60 min, and then at 24, 48, and 72 h from patch removal. Draize’s dermal irritation scoring model was adopted for dermal assessment, as per OECD recommendation shown in Table 5.

### 3.15. Stability Study of the Selected AzA Formula

A preliminary stability study was conducted on MCP116 to determine the physicochemical stability of storage in different environments. A fresh sample of MCP116 was prepared as outlined previously, and the amount was divided into nine tubes; three tubes were sealed and stored in a fridge at 4 °C. Another three tubes were sealed and stored in an oven at 40 °C, and the last three tubes were sealed and stored at room temperature conditions at 20–25 °C. Each sample was wrapped with aluminum foil to mitigate against possible photosensitivity. The samples were tested at regular intervals (10 days, 20 days, and 30 days), investigating for changes in both organoleptic properties and drug content.

## 4. Conclusions

The choline-based DESs are recently gaining great focus for many researchers in the field of transdermal, topical and oral drug delivery. The work outlined herein is an application of choline-based DESs as a topical delivery system of azelaic acid. The results obtained indicate the suitability of the application of DESs technology as a solubilizing and drug-delivery vehicle of azelaic acid for acne treatment. The method of preparation is simple with high loading capacity, good stability, and higher permeability than the commercial product. The DES itself showed antibacterial activity and synergetic antibacterial effect with AzA. Such synergistic effect due to DES may allow for dose reduction and so the study provides an added value for DES application as solubilizer and drug delivery vehicle. The high reported viscosity of the DES was simply modified by the addition of PEG 400, which showed excellent miscibility with the tested DES system. The DES and the PEG aid in the enhancement of drug diffusion through the tested membrane.

## Figures and Tables

**Figure 1 molecules-28-06927-f001:**
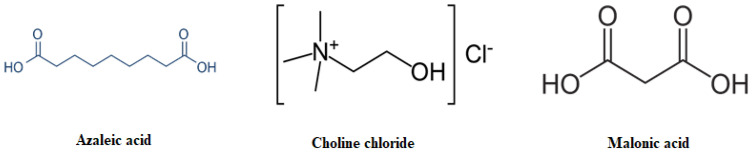
Chemical structure of Azelaic acid, Malonic acid, and Choline chloride.

**Figure 2 molecules-28-06927-f002:**
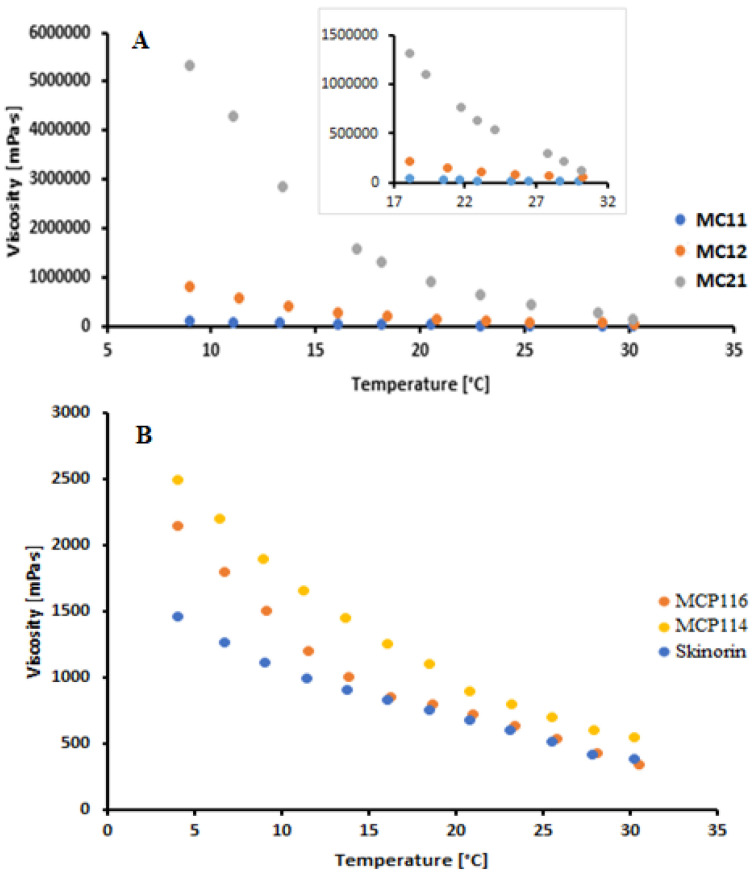
Rheogram of the binary deep eutectic system from malonic acid and choline chloride at different molar ratios (**A**) and malonic acid and choline chloride with PEG400 (**B**) at temperatures from 8 °C to 32 °C.

**Figure 3 molecules-28-06927-f003:**
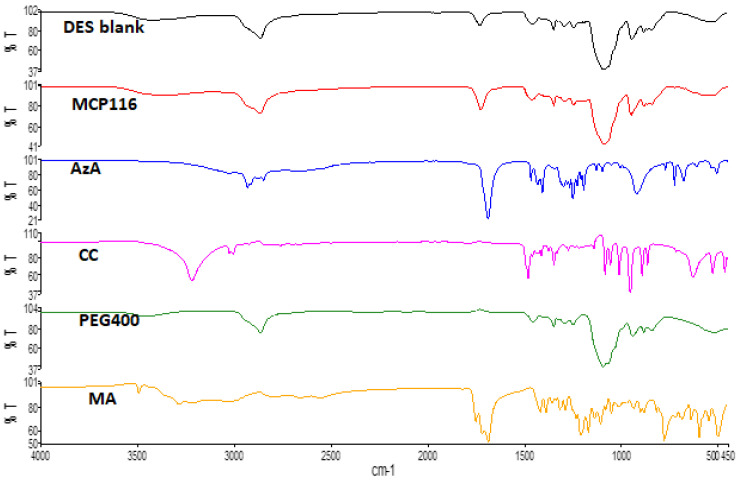
FTIR spectra of MA, CC, AzA, PEG, and the optimal candidate formulation MCP116.

**Figure 4 molecules-28-06927-f004:**
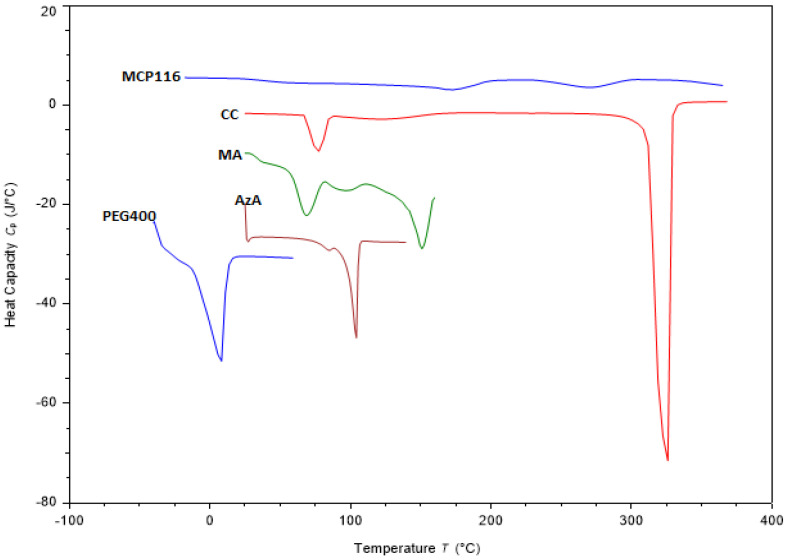
DSC thermogram of choline chloride (CC), malonic acid (MA), Azelaic acid (AzA, PEG400, and the eutectic formulation MCP116, with AzA.

**Figure 5 molecules-28-06927-f005:**
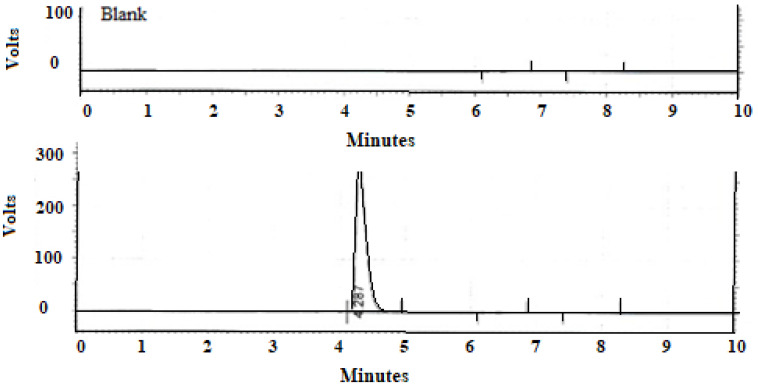
Chromatogram of blank formulation and formulation with Azelaic acid.

**Figure 6 molecules-28-06927-f006:**
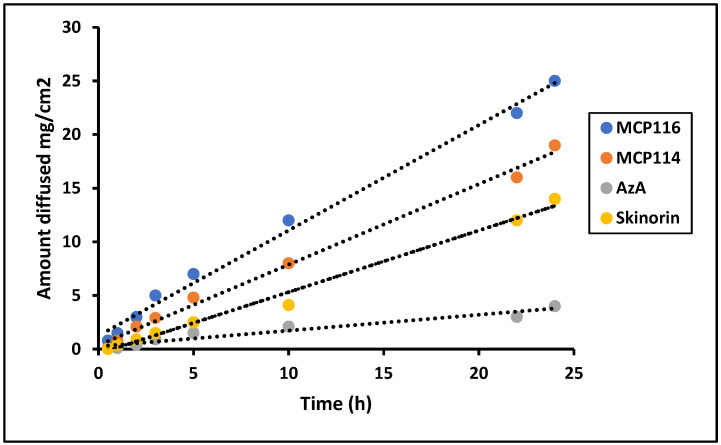
Diffusion study of Formulation MCP116, MCP114, AzA suspension and commercial product (Skinorin) through polyamide membrane, temperature 37 °C.

**Table 1 molecules-28-06927-t001:** The solubility of AZA, pH of the DESs, partition coefficient of AzA in the DESs, viscosity, spreadability and contact angle of the DESs.

DESs	AzA Solubility mg/g IL Mean ± SD	pH ± SD	Log *p*	Viscosity (mPa.s) ± SD at 30 °C	Spreadability Cm (Mean ± SD)	Contact Angle (Θ)
MC11	NM	NM	NM	15,872 ± 20.1	2.91 ± 0.13	NM
MC12	NM	NM	NM	51,950 ± 25.1	2.91 ± 0.13	NM
MC21	NM	NM	NM	403,000 ± 25.1	2.91 ± 0.13	NM
MCP111	156.5 ± 7.1	3.01 ± 0.01	2.12	1504 ± 12.1	5.2 ± 0.01	75 ± 3.4
MCP112	172.25 ± 8.2	3.11 ± 0.02	2.40	1076 ± 7.2	8.0 ± 0.12	70 ± 2.9
MCP114	178.0 ± 6.8	3.00 ± 0.01	2.65	721 ± 5.1	9.2 ± 0.21	65 ± 2.4
MCP116	194.1 ± 10.1	3.13 ± 0.01	2.91	340 ± 7.1	11.4 ± 0.31	53 ± 2.1
MCP121	92.0 ± 7.6	2.60 ± 0.01	2.12	5120 ± 19.1	3.01 ± 0.13	79 ± 3.4
MCP122	130.5 ± 8.9	2.61 ± 0.02	2.21	4440 ± 10.2	5.5 ± 0.16	76 ± 2.9
MCP124	145.5 ± 8.1	2.62 ± 0.01	2.21	822 ± 9.2	7.7 ± 0.12	72 ± 2.4
MCP126	152.6 ± 10.1	2.62 ± 0.015	2.45	646 ± 7.1	8.3 ± 0.21	69 ± 2.1
Skinorin^®^	-	-	-	380 ± 5.1	7.3 ± 0.11	-
Azelaic acid	≈0.24 g/100 g	-	2.01	-	-	-

NM: the reading was not possible to perform due to the high viscosity.

**Table 2 molecules-28-06927-t002:** Inhibition zone diameters of the DES and Skinorine^®^ against *C. acnes* at the end of 72 h (*n* = 3; * *p* < 0.05 for MCP116 vs. Skinorine^®^; *p* < 0.05 for MCP116, with AzA vs. MCP116, without AzA).

Formulation	Inhibition Zone (mm) ± SD
MCP116, with AzA	28.62 ± 0.85 *
MCP116, without AzA	09.03 ± 0.90
Skinorine^®^	21.50 ± 0.81

**Table 3 molecules-28-06927-t003:** Dermal responses observed in individual rabbits for the formulation MCP116.

**Erythema**					
Rabbit	Evaluation after removal of the test substance				
	0 min	60 min	24 h	48 h	72 h
1 (Initial)	0	0	0	0	0
2 (Confirmatory)	0	0	0	0	0
**Edema**					
Rabbit	Evaluation after removal of test substance				
	0 min	60 min	24 h	48 h	72 h
1 (Initial)	0	0	0	0	0
2 (Confirmatory)	0	0	0	0	0

**Table 4 molecules-28-06927-t004:** Combinations of the composition of DESs.

Code	Components	Ratio (MA:CC: PEG (MCP) Respectively)
MC11	MA: CC	1:1
MC12	MA: CC	1:2
MC21	MA: CC	2:1
MCP111	MA:CC: PEG400	1:1:1
MCP112	MA:CC: PEG400	1:1:2
MCP114	MA:CC: PEG400	1:1:4
MCP116	MA:CC: PEG400	1:1:6
MCP121	MA:CC: PEG400	1:2:1
MCP122	MA:CC: PEG400	1:2:2
MCP124	MA:CC: PEG400	1:2:4
MCP126	MA:CC: PEG400	1:2:6

**Table 5 molecules-28-06927-t005:** Draize’s dermal irritation scoring model adopted for dermal assessment, as per OECD recommendation.

Erythema and Eschar Formation	Value	Edema Formation	Value
No erythema	0	No edema	0
Very slight erythema (barely perceptible)	1	Very slight edema (barely perceptible)	1
Well-defined erythema	2	Slight edema (edges of area well defined by definite raising)	2
Moderate to severe erythema	3	Moderate edema (raised approximately 1 mm)	3

## Data Availability

No data is confidential.

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
