# Peer review of "The Formulation and Evaluation of Deep Eutectic Vehicles for the Topical Delivery of Azelaic Acid for Acne Treatment"

_molecules, 2023, doi:10.3390/molecules28196927_

Round 1

Reviewer 1 Report

Dear Author,

The topic of proposed paper „Formulation and Evaluation of Deep Eutectic Vehicles for the Topical Delivery of Azelaic Acid for Acne Treatment“ is in the line with the current research on therapeutic deep eutectic solvents. It is very interesting from applicative point of view, since different ways to obtaine improved formulations of AzA are investigated and according to a literature survey this the first time that DES-API approach was investigated.

Methodologicaly this investigation is well planed and done, and obtained results are mostly presented good and clearly.

Specific comments to the authors:

·         You have written why you choose choline chloride (CC), malonic acid (MA), and PEG 400, and I understand your reasons for it, but to improve discussion please check are does compounds approved for topical application and add comment on that.

·         For the sake of comparison, I suggest to put value of AzA solubility in water (line 127) also in Table 3.

·         Do you have results of rheology for the MCP116, which showed the highest solubility of AzA, to include it in Fig. 2. That is the chosen formulation so that result should be also included.

·          

Small mistakes:

·         The Figure 1 is missing? The first Figure in the manuscript is Fig. 2. Check it and correct it accordingly! And renumber the following Figures.

·         You have to different results under the same name: Figure 5, i.e. you have two Figures 5, but not Figure 4! Check it and correct it accordingly in the manuscript text and Figure capitations.

·         Sentences in Lines 250-252 are probably from the Manuscript template and therefore redundant so deleted it.

·         The end of the sentence “…is miscible with the tested DES system.” In the line 415-416 seems excessive.

Author Response

Response to Reviewer # 1

  1.  You have written why you chose choline chloride (CC), malonic acid (MA), and PEG 400, and I understand your reasons for it, but to improve the discussion please check are does compounds approved for topical application and add a comment on that.

Response: The following paragraph has been added to the introduction (Moreover, the suggested components of the DES utilized in the current study have been reported as suitable vehicles for topical applications and are generally recognized as safe (GRAS) when used in accordance with good manufacturing practices [30].

In addition, the following statement has been added in response to reviewer 2 comment on the introduction paragraph

To the best of our knowledge, the pharmaceutical application of a DES made from MA and CC as a solubilizing and drug delivery vehicle for AzA was not reported before. Therefore, the present work describes an investigation into the feasibility of the use of a DES formulation consisting of MA, CC, and PEG 400 as a vehicle for the intradermal delivery of AzA to treat acne. The prepared DES will be evaluated in terms of viscosity, spreadability, pH, partition coefficient, contact angle, and solubilization power.

Moreover, the following paragraph has been added to the manuscript that may answer your valid concern

The pH values of the formulations in the range of 2.6 to 3.11 are generally lower than the typical pH range of healthy skin, which falls between 4 to 6. Lower pH values can make formulations more acidic, which could potentially lead to skin irritation, dryness, or disrupt the skin's natural pH balance. However, the suitability of these pH values depends on various factors, including the specific ingredients in the formulation, the intended use (e.g., as a spot treatment or all-over application), and individual skin sensitivities. Since the current formulation is intended for short-term and localized use (acne treatment), a slightly lower pH might be acceptable and justifiable as it can help solubilize AzA, and enhance its effectiveness and intradermal penetration. Furthermore, it is worth noting that the components comprising the formulation, namely CC, MA, and PEG400, are generally considered "green" components. They have a well-documented history in skin and cosmetic formulations, demonstrating acceptable levels of skin irritation and tolerability. These attributes can further support the justification for the pH range chosen in the formulation.

  • 2. For the sake of comparison, I suggest putting the value of AzA solubility in water (line 127) in Table 3.

Response: The solubility value of AzA is already been mentioned in the manuscript (line 128) in the following paragraph( Among the tested DESs, MCP116 showed the highest solubility with about 80 times the solubility of the drug in water (AzA solubility in water is 2.4 mg/mL) [14].). It has been added to Table 3

  • 3.   Do you have the results of rheology for the MCP116, which showed the highest solubility of AzA, to include it in Fig. 2. That is the chosen formulation so that result should be also included.

Response: The results of the viscosity of formulation MCP116 in Figure 2B

Small mistakes:

  • 1.  The Figure 1 is missing? The first Figure in the manuscript is Fig. 2. Check it and correct it accordingly! And renumber the following Figures.

Response: Figure 1 has been added to the manuscript which includes the structure of CC, MA, and AZA and no need to adjust the sequence of references

  • 2.  You have to different results under the same name: Figure 5, i.e. you have two Figures 5, but not Figure 4! Check it and correct it accordingly in the manuscript text and Figure capitations.

Response: By mistake, the discussion of the FTIR results mentioned two Figures 4 and 5. In fact, all the results are included in one figure only namely Figure 4 so figure 5 has been changed to figure 4(line 185) in the text, and no need to adjust the number sequence of the figures in the text.

  • 2. Sentences in Lines 250-252 are probably from the Manuscript template and therefore redundant so deleted it.

Response: Yes. The two lines have been deleted from the text.

  • The end of the sentence “…is miscible with the tested DES system.” In line 415-416 seems excessive.

Response: Yes. The sentence (with the DES is miscible) has been deleted from the text.

Reviewer 2 Report

1. Abstract:  Page 1, line 32. the sign of degree must be superscript.

2. Introduction: the authors have mentioned all the background of the study. But no aim and objective have been mentioned in the last paragraph of introduction.

3. There is an issue in terms of plagiarism. it must be less than 15 %.

4. For supporting of the results of microbiology and skin irritation study, the authors must provide images to prove the effectiveness and no toxicity.

5. Page 4 table 3, the pH values of formulations ranges from 2.6 to 3.11. are these values are suitable for skin formulations, as skin pH ranges between 4 to 6. need to justify.

6. Lot of studies are available regarding the acne treatment of AzA. The authors did not highlight any novelty and results compared with previously published reports.

7. Regarding the HPLC method, the authors did not provide any chromatogram of blank formulation indicating no interferance of choline chloride and malonic acid.

8. Typographical mistakes throughout the manuscript. it must be corrected. For example, after full stop, space must be given. Abbreviations must be used uniformly etc.

Author Response

Responses to Reviewer # 2

  1. Abstract:  Page 1, line 32. the sign of degree must be superscript.

Response: Corrected

  1. Introduction: The authors have mentioned all the background of the study. However, no aim and objective has been mentioned in the last paragraph of the introduction.

Response: The following aim and objective of the study has been added at the end of the introduction. To the best of our knowledge, the pharmaceutical application of a DES made from MA and CC as a solubilizing and drug delivery vehicle for AzA was not reported before. Therefore, the present work describes an investigation into the feasibility of the use of a DES formulation consisting of MA, CC, and PEG 400 as a vehicle for the intradermal delivery of AzA to treat acne. The prepared DES will be evaluated in terms of viscosity, spreadability, pH, partition coefficient, contact angle, and solubilization power.

  1. There is an issue in terms of plagiarism. it must be less than 15 %.

Response: This issue has been taken into consideration and the plagiarism is now within the accepted limits

  1. To support the results of the microbiology and skin irritation study, the authors must provide images to prove the effectiveness and no toxicity.

Response: Since the irritation test scored zero as shown in Table 5, images could not add any further interpretation of the obtained results

  1. On Page 4 table 3, the pH values of formulations range from 2.6 to 3.11 are these values suitable for skin formulations, as skin pH ranges between 4 to 6 need to justify.

Response: The following paragraph has been added to the text. The pH values of the formulations in the range of 2.6 to 3.11 are generally lower than the typical pH range of healthy skin, which falls between 4 to 6. Lower pH values can make formulations more acidic, which could potentially lead to skin irritation, dryness, or disrupt the skin's natural pH balance. However, the suitability of these pH values depends on various factors, including the specific ingredients in the formulation, the intended use (e.g., as a spot treatment or all-over application), and individual skin sensitivities. Since the current formulation is intended for short-term and localized use (acne treatment), a slightly lower pH might be acceptable and justifiable as it can help solubilize AzA, and enhance its effectiveness and intradermal penetration. Furthermore, it is worth noting that the components comprising the formulation, namely CC, MA, and PEG400, are generally considered "green" components. They have a well-documented history in skin and cosmetic formulations, demonstrating acceptable levels of skin irritation and tolerability. These attributes can further support the justification for the pH range chosen in the formulation.

  1. A lot of studies are available regarding the acne treatment of AzA. The authors did not highlight any novelty and results compared with previously published reports.

Response: the following paragraph has been added in response to this valuable comment

In summary, this formulation represents a pioneering approach to AzA delivery for acne treatment, utilizing DES. It offers several advantages, including safety, environmental friendliness (as it is free of organic solvents), and simplicity in preparation. Anti-acne studies demonstrated that the DES formulation exhibited inhibition zones comparable to those of the commercially available cream and previously reported formulations [46-48].

  • Apriani, E. F., Rosana, Y., & Iskandarsyah, I. Formulation, characterization, and in vitro testing of azelaic acid ethosome-based cream against Propionibacterium acnes for the treatment of acne. J. Adv. Pharm. Technol. Res. 2.019, 10(2), 75.‏ https://doi.org/10.4103%2Fjaptr.JAPTR_289_18
  1. Arpa, M. D., Seçen, İ. M., Erim, Ü. C., Hoş, A., & Üstündağ Okur, N. Azelaic acid loaded chitosan and HPMC based hydrogels for treatment of acne: formulation, characterization, in vitro-ex vivo evaluation.Pharm Dev Technol, 2.22, 27(3), 268-281.‏ https://doi.org/10.1080/10837450.2022.2038620
  2. Ghasemiyeh, P., Mohammadi-Samani, S., Noorizadeh, K., Zadmehr, O., Rasekh, S., Mohammadi-Samani, S., & Dehghan, D. Novel topical drug delivery systems in acne management: Molecular mechanisms and role of targeted delivery systems for better therapeutic outcomes. J Drug Deliv Sci Technol, 2022, ‏ https://doi.org/10.1016/j.jddst.2022.103595

  1. Regarding the HPLC method, the authors did not provide any chromatogram of blank formulation indicating no interference of choline chloride and malonic acid.

Response: A blank formulation chromatogram has been added to Figure 5

  1. Typographical mistakes throughout the manuscript. it must be corrected. For example, after full stop, space must be given. Abbreviations must be used uniformly etc..

Response: The entire manuscript has undergone revision to address language, grammar, and typographical errors.

Round 2

Reviewer 2 Report

The authors have done all the recommended changes. It can now be acceptable in present form.